# Effect of Fine Particulate Matter Exposure on Liver Enzymes: A Systematic Review and Meta-Analysis

**DOI:** 10.3390/ijerph20042803

**Published:** 2023-02-04

**Authors:** Ling Pan, Jing Sui, Ying Xu, Qun Zhao, Yinyin Cai, Guiju Sun, Hui Xia

**Affiliations:** 1Research Institute for Environment and Health, Nanjing University of Information Science and Technology, Nanjing 210044, China; 2Key Laboratory of Environmental Medicine Engineering, Ministry of Education, School of Public Health, Southeast University, Nanjing 210009, China; 3Institute of Atmospheric Environmental Economics, Nanjing University of Information Science and Technology, Nanjing 210044, China

**Keywords:** fine particulate matter, liver diseases, liver enzyme, meta-analysis

## Abstract

Although previous studies have presented that fine particulate matter (PM2.5) regulates liver enzyme levels in the development of liver diseases, the evidence regarding the relationship between PM2.5 exposure and liver enzyme is not robust. We further aimed to conduct a systematic review and meta-analysis of observational studies to summarize the recent evidence on the effects of PM2.5 on liver enzyme in humans. In the meta-analysis, we retrieved online databases including PubMed and Web of Science database from 1982 up to 2022. A random-effects model was applied to evaluate the correlation between PM2.5 and liver enzyme level. A total of 10 studies fulfilled the inclusion criteria, including five prospective cohort studies, two cross-sectional studies, two longitudinal studies, and one time-series analysis. Each 10 μg/m^3^ increase in PM2.5 concentration was significantly correlated with a 4.45% increase in alanine aminotransferase (ALT) level (95% CI: 0.51–8.38%, *p* = 0.03), a 3.99% increase in aspartate transferase (AST) level (95% CI: 0.88–7.10%, *p* = 0.01), and a 2.91% increase in gamma-glutamyl transferase (GGT) level (95% CI: 1.18–4.64%, *p* < 0.001), but this significant association was not observed in alkaline phosphatase (ALP). Subgroup analysis revealed that PM2.5 has a significant correlation with ALT (5.07%, 95% CI: 0.81–9.33%), AST (4.11%, 95% CI: 0.74–7.48%), and GGT (2.74%, 95% CI: 1.09–4.38%) in Asia. Our meta-analysis showed that increments in PM2.5 exposure were significantly associated with a higher level of ALT, AST, and GGT. In addition, investigations into liver enzyme subtypes and specific chemical components of PM2.5 are important directions for future research.

## 1. Introduction

Fine particulate matter (PM2.5) represents fine inhalable particles, with diameters of generally 2.5 μm and smaller [1]. Because of the small diameters of PM2.5, it is easily inhaled into the lungs and deposited in the alveoli and even the bloodstream, resulting in an inflammatory response [2]. PM2.5 invades the alveoli, enters the blood circulation, and reaches different organs thanks to its small particle size, consequently endangering human health [3,4].

Air pollutants have been found to be involved in various oxidative stress mechanisms, which leads to inflammation and affects the liver and other organs [5,6]. Increasing evidence has shown that liver enzyme levels were significantly related to exposure to PM2.5 [7,8]. Epidemiologic studies found that PM2.5 exposure caused the change in alkaline phosphatase (ALP) level in the blood of obese children after bariatric surgery for treatment [9]. The relationship between PM2.5 and various disease was partly mediated by liver enzymes, including liver diseases [10,11].

Liver enzymes are important parameters in the liver function examination [12], mainly including alanine aminotransferase (ALT), aspartate transferase (AST), ALP, and gamma-glutamyl transferase (GGT) [13]. So far, accumulating studies have shown that people exposed to air pollutants may experience liver injury and increased liver enzyme levels [5,14]. Abnormal elevation of ALT and AST levels may promote acute or chronic inflammation of the liver and other chronic liver diseases [15,16].

Current meta-analyses primarily focus on the incidence and mortality of liver diseases and PM2.5 [17,18]. In another meta-analyses, a PM2.5 increase of 10 μg/m^3^ was significantly correlated with liver cancer (HR:1.22, 95% CI: 1.14–1.30, *p* < 0.05) [19]. Pritchett et al. [20] reported a robust correlation between PM2.5 exposure and hepatocarcinogenesis. Some epidemiological studies have confirmed the positive relationship between the concentration of PM2.5 and the level of liver enzymes [21]. Kyung Nam Kim et al. reported that every increase of one standard inter-quartile range (IQR = 13.2 µg/m^3^) of PM2.5 increased AST (increased by 3.0% (95% CI: 0.9–5.1%), ALT (increased by 3.2% (95% CI: 0.3–6.2%)), and γ- GTP (increased by 5.0% (95% CI: 1.5–8.7%)) levels [5]. In a cross-sectional study, every rise of 10 μg/m^3^ in PM2.5 increased AST levels by 0.02% (95% CI: −0.04–0.08%), ALT levels by 0.61% (95% CI: 0.51–0.70%), and GGT levels by 1.60% (95% CI: 1.50–1.70%) [22]. Therefore, our present meta-analysis aims to investigate the association between PM2.5 and the liver enzyme level based on observational epidemiological studies to assess the impact of PM2.5 on liver disease.

## 2. Materials and Methods

### 2.1. Data Sources and Searches

We systematically searched the online database of PubMed and Web of Science for PM2.5 induced air pollution and liver enzymes from 1982 to November 2022. The search terms were as follows: “PM2.5”, “Air Pollution”, “Fine Particulate Matter”, “Particulate Matter”, “Liver Enzymes”, “Biological Marker”, “ALP”, “Alkaline Phosphatase”, “ALT”, “Alanine Transaminase”, “Glutamic-Alanine Transaminase”, “AST”, “Aspartate Aminotransferase”, “Aspartate Transaminase”, “GGT”, “Glutamyl Transpeptidase”, and “gamma-Glutamyl Transferase”. Only publications in the English language were included in the analysis. We also referred to the reference list of the original documents to determine other relevant data.

### 2.2. Study Selection and Eligibility

Our selection criteria were (1) observational epidemiological study; (2) correlation between PM2.5 concentration and liver enzyme level; (3) reported results measuring the liver enzyme level, correlations, and 95% CI; and (4) data of PM2.5 exposure levels were collected from monitoring station, horizontal–vertical locations, or satellite. When two or more analysts shared data or have the same participants, we conducted a more comprehensive analysis. When searching for articles, we excluded the following contents: (1) case reports, letters, and reviews; (2) studies without PM2.5 increment; (3) relevant data of animals or cells; (4) studies without subdividing particle types; and (5) studies without liver enzyme level or relevant data. Two independent investigators (L.P. and Y.X.) evaluated the eligibility of the study according to the inclusion criteria. The differences involved in the evaluation were discussed and resolved by the third author (J.S.).

### 2.3. Data Extraction

Two reviewers (L.P. and Y.X.) independently extracted the year of publication, name of the first author, research type, research location, number of participants, gender of participants, liver enzyme type, adjusted variables, and adjusted 95% CI from the qualified articles selected.

### 2.4. Literature Quality Assessment

We evaluated the quality of eligible literature works and scored all publications from 0 to 9 on the Newcastle–Ottawa Scale (NOS) [23]. Studies awarded higher scores than the mean score were considered as high-quality studies.

### 2.5. Statistical Analyses

Here, 10 μg/m^3^ is defined as the standardized increment of PM2.5. Regression coefficients were transformed to percent changes according to the equation [β× 10 ÷ M] × 100% when β was not log-transformed. In this equation, β represents the regression coefficient and M is the arithmetic mean of the liver enzyme level. Natural log-transformed data were anti-log transformed. We use Cochran’s Q and I^2^ to test the heterogeneity in the study. I^2^ from 0 to 100% means that the research has changed from no observation to maximum heterogeneity. I^2^ > 50% showed serious heterogeneity. We used the Begg funnel plot and Egger test to calculate the publication bias. We used Stata statistical software 11.0 for all statistical analyses.

## 3. Results

### 3.1. Eligible Studies

The article filtering process is shown in Figure 1. After the initial literature search, we identified 16,183 studies. After excluding 885 repetitive literature works, 15,298 studies were preliminarily screened. After that, we excluded 9536 articles on animals and cells. According to the criteria for induction and exclusion, and the review of literature titles and abstracts, 5601 articles did not meet the inclusion criteria. Finally, we reviewed the remaining 161 articles and found that 151 articles did not qualify. The reasons were as follows: articles that did not report percent change, regression coefficient, or fold change (*n* = 108); no circulating biomarkers measure (*n* = 18); no PM2.5 data (*n* = 10); and no PM2.5-increment data (*n* = 15). The remaining ten studies, including five cohort studies [3,5,9,10,11], two cross-sectional studies [14,22], two longitudinal studies [24,25], and one time-series analysis [26], were included in the final analysis.

### 3.2. Characteristics of Studies Included in the Meta Analysis

The characteristics of 10 studies included in the final meta-analysis are shown in Table 1. All studies were published between 2013 and 2022, including more than 14 million participants. The relationship between PM2.5 and liver enzymes including ALP, ALT, AST, and GCT levels were reported in the above studies. According to the continental division, of 10 studies, 8 were conducted in Asia, 1 in Europe, and 1 in North America. The number of participants in the study ranged from less than 100 to more than 13 million. According to the NOS score, the score range of the nine studies was 6–9 and the average score was 8.1. According to the criteria that the NOS score was ≥8, four studies were of high quality.

### 3.3. Overall Meta Estimates and Publication Bias

Heterogeneity was observed among studies investigating ALP (I^2^ = 98.26%, *p* = 0.07), ALT (I^2^ = 99.47%, *p* = 0.03), AST (I^2^ = 99.34%, *p* = 0.01), and GGT (I^2^ = 49.14%, *p* < 0.001).

Figure 2 shows the effects of an elevated PM2.5 concentration on the levels of four liver enzymes. In the random-effects model, no statistical significance between PM2.5 and liver ALP was observed. In general, however, every 10 μg/m^3^ increase in PM2.5 exposure was related to a 6.28% (95% CI: −0.56–13.12%, *p* = 0.07) increase in the liver ALP level. There was significant correlation between PM2.5 concentration and liver ALT, AST, and GGT levels. PM2.5 rose every 10 μg/m^3^, ALT level increased by 4.45% (95% CI: 0.51–8.38%, *p* = 0.03), AST level increased by 3.99% (95% CI: 0.88–7.10%, *p* = 0.01), and GGT level increased by 2.91% (95% CI: 1.18–4.64%, *p* < 0.001). The result of the Egger’s test for asymmetry showed *p* > 0.05, indicating no publication bias in all types of liver enzymes.

### 3.4. Subgroup Analysis of PM2.5 on Changes in Liver Enzyme Levels

The subgroup analysis of PM2.5 concentration on changes in liver enzyme level is shown in Table 2. When exposed to PM2.5, sample size ≥ 1000 had a stronger significant association with the GGT level (4.73%, 95%CI: 0.85–8.61%) than sample size < 1000 (2.25%, 95%CI: 0.30–4.20%). However, sample size ≥ 1000 had no significant association with AST (3.40%, 95% CI: −0.28–7.09%), nor did sample size < 1000 (5.84%, 95% CI: −1.56–13.25%). Regarding study design, PM2.5 had a lower significant effect on AST level (3.68%, 95%CI: 1.15–6.22%) in the prospective cohort study than others (4.30%, 95%CI: 0.50–9.11%). Whether a prospective cohort study (ALT:6.28%, 95%CI: −3.34–15.90%) or another study (ALT: 4.18%, 95%CI: −2.52–10.88%), PM2.5 had no significant association with ALT and GGT. In addition, the subgroup analysis of age revealed that age ≥ 60 (ALT: 6.79%, 95%CI: −1.66–15.23%; AST:6.68%, 95%CI: −1.92–15.27%) and age < 60 (ALT: 2.02%, 95%CI: −1.56–5.61%; AST:4.47%, 95%CI: −4.87–14.34%) had no significant correlation with ALT level and AST level in per IQR of PM2.5.

## 4. Discussion

At present, studies have pointed out that PM2.5 pollution is one of the main reasons for the increase in the global incidence rate [27,28]. An increase in PM2.5 concentration could cause damage to the liver [29]. Our meta-analysis involved 10 studies and more than 14 million people in five countries. In our present meta-analysis, we found that PM2.5 exposure was positively correlated with liver enzyme levels, including AST, ALT, or GGT levels. However, there was no significant correlation between ALP level and PM2.5 exposure concentration.

PM2.5 exposure not only promoted the incidence rate and mortality of respiratory diseases, but also had a significant impact on digestive diseases [30,31]. Liver enzymes were reliable markers to judge whether the liver was damaged [32]. Studies have demonstrated that liver enzymes played an important logical intermediary role in the association between PM2.5 and some digestive diseases [11]. In our meta-analysis, liver ALT, AST, and GGT concentrations were found to be positively correlated with PM2.5 exposure increase. Every increment of 10 μg/m^3^ in PM2.5 was associated with a 4.45% increase in ALT (95% CI: 0.51–8.38%, *p* = 0.03), 3.99% increase in AST (95% CI: 0.88–7.10%, *p* = 0.01), and 2.91% increase in GGT (95% CI: 1.18–4.64%, *p* < 0.001), respectively.

We tried to propose some mechanisms to explain the relationship between PM2.5 and liver enzyme levels. The first mechanism involved oxidative stress causing liver function damage. The increase in liver enzymes could indicate that PM2.5 causes liver system diseases caused by oxidative stress [33,34]. Exposure to PM2.5 will lead to the production of reactive oxygen species (ROS) and an increase in oxidative stress in the liver, increase the level of liver enzymes, and affect the normal metabolism of the liver [35,36,37]. It has been pointed out that liver serum biomarkers (ALT and AST) induced an increase in free radical levels, which may cause oxidative stress [38]. PM2.5 exposure promoted the expression of Nrf-2 and Nrf-2 regulated antioxidant genes, leading to the imbalance of oxidative stress and redox in the liver [39].

Another important mechanism is inflammation. The induction of PM2.5 will aggravate liver inflammation and lead to abnormal liver function [40]. As an important basis for judging liver injury, changes in liver enzyme levels were highly susceptible to inflammation [41]. Tumor necrosis factor-α (TNF-α), interleukin-6 (IL-6), and other inflammatory factors activated by PM2.5 were accompanied by an increase in liver enzyme level [28,42,43]. Wei et al. [44] reported that PM2.5 would increase TNF-α. The production of IL-6 and IL-6 induced liver inflammation in mice. At the same time, it was accompanied by an increase in ALT, AST, and GGT levels, leading to a severe inflammatory liver reaction in mice [44].

The advantage of our meta-analysis is that it covers more than 14 million participants and analyzes the impact of PM2.5 exposure on different types of liver enzymes. At present, meta-analyses on air pollution and liver focused on analyzing liver diseases caused by air pollution. In this study, we focused on the mediating role of liver enzymes in various liver system diseases. As far as we know, our meta-analysis is the first to explore the relationship between PM2.5 and various liver enzyme levels, which will also be our greatest advantage.

Meanwhile, our study has some limitations. First, few articles focused on the relationship between PM2.5 exposure and liver enzyme level, and only 10 articles were included in the analysis. Therefore, there may be heterogeneity caused by differences between individuals and different observation characteristics, which may affect the results. Second, the number of studies in different groups included in the subgroup analysis was small, and the results may be biased. Finally, because the increase in liver enzymes may be caused by reasons other than liver cell damage, it is necessary to further explore biomarkers with high liver specificity.

## 5. Conclusions

Our results indicated that there was a close relationship between air pollution and liver enzyme levels. When the exposure concentration of PM2.5 increases, the level of human liver enzymes will rise accordingly. Although specific liver enzyme data were limited, our data showed that PM2.5 exposure will increase the levels of ALT, AST, and GGT, causing liver damage. Although increasingly more studies have focused on the impact of air pollution on the liver system, there is a lack of research focusing on air pollution and liver enzymes. Up to now, no meta-analysis of PM2.5 and liver enzymes has been conducted, and our research attempts to propose two mechanisms to explain them. As one of the important mediators of air pollution affecting liver disease, the relationships between other air pollutants and liver enzymes need to be further studied.

## Figures and Tables

**Figure 1 ijerph-20-02803-f001:**
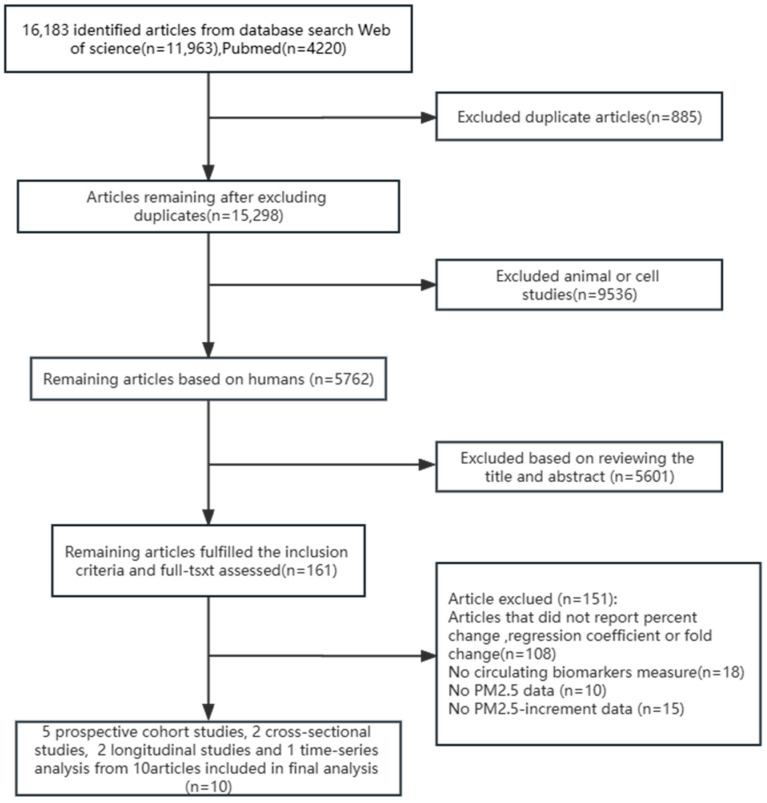
Flow diagram for the identification of relevant studies.

**Figure 2 ijerph-20-02803-f002:**
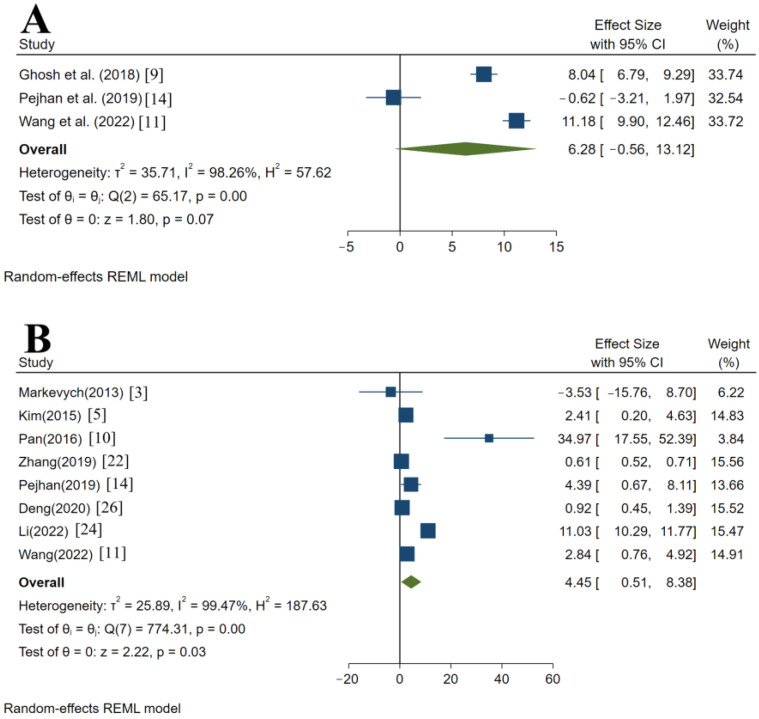
Association of PM2.5 exposure with levels of ALP (**A**), ALT (**B**), AST (**C**), and GGT (**D**) according to a random-effects meta-analysis. CI, confidence interval (95% CI are for a 10 μg/m^3^ increase in PM2.5); ALP, alkaline phosphatase; ALT, alanine aminotransferase; AST, aspartate transferase; GGT, gamma-glutamyl transferase.

**Table 1 ijerph-20-02803-t001:** General characteristics of the included studies.

Studies	Study Design	Location	Years Enrolled	Age Range (Years)	Gender	Sample Size	Liver Enzyme Types	Adjustment Variables	NOS
Markevych et al. [3] (2013)	Prospective Cohort Study	Europe	2004–2009	31–85	Male/female	5892	GGT, AST, and ALT	Socioeconomic, lifestyle, and clinical covariates.	9
Kim et al. [5] (2015)	Prospective Cohort Study	Korea	2008–2010	≥60	Male/female	545	GGT, AST, and ALT	Age, sex, smoking status, mean temperature, dew point, season, body mass index, alcohol consumption, and amount of exercise.	9
Pan et al. [10] (2016)	Prospective Cohort Study	Taiwan	1991–2009	30–65	Male/female	22,062	ALT	Age, sex, alcohol consumption, smoking, HBsAg serostatus, anti-HCV serostatus, and county at study entry.	8
Ghosh et al. [9] (2018)	Prospective Cohort Study	USA	2005–2014	<18	Male/female	75	ALP	Sex, age, race/ethnicity and weight at the time of surgery.	6
Pejhan et al. [14] (2019)	A Cross-Sectional Study	Iran	2018	average 27.7 ± 5.4	Female	150	AST, ALT, ALP, and GGT	Age of mother, BMI of mother before pregnancy, number of pregnancies, gestational age, percent of illiterate per census tract, percent of unemployment per census tract, paternal education, maternal education, income, tobacco exposure at home, newborn sex, newborn’s BMI, paternal education, exposure to environmental tobacco smoke, car ownership, home ownership, use the hood during cooking, time of cooking in each day during pregnancy, and time of exposure to cigarette smoke at home during pregnancy.	8
Zhang et al. [22] (2019)	Cross-Sectional Study	Taiwan	2001–2014	average 40.1 ± 13.1	Male/female	351,582	AST, ALT, and GGT	Age, sex, educational level, smoking, alcohol drinking, leisure-time physical activity, occupational exposure to dust and organic solvent, season, body mass index, hypertension, diabetes, hyperlipidaemia, self-reported cardiovascular disease or stroke, self-reported cancer and self-reported liver disease (hepatitis and cirrhosis).	9
Deng et al. [26] (2020)	Time-Series Analysis	China	2014–2016	all ages	Male/female	13,045,629	AST and ALT	Time trends, weather conditions (temperature and humidity), days of the week, and the effects of other air pollutants.	8
Wang et al. [11] (2022)	Prospective Cohort Study	China	2018–2021	30–79	Male/female	7963	ALT, AST, ALP, and GGT	Demographic characteristics, including age, sex, annual household income, ethnic group, residential type, lifestyle behaviors (smoking status, secondhand smoke status, alcohol consumption, indoor pollution, physical activity, and Mediterranean diet score), and environmental factors (season and nitrogen dioxide).	9
Li et al. [24] (2022)	Longitudinal Study	China	2018–2020	65.0–120.4	Male/female	318,911	AST and ALT	Sex, age, race, educational attainment, cigarette smoking, alcohol consumption, physical activity, BMI categories, abdominal obesity, hypertension, diabetes, dyslipidemia, year, and season.	8
Hu et al. [25] (2022)	Longitudinal Study	China	2013–2020	18–99	Male/female	247,640	AST	Age, gender, body mass index, smoke status, cardiometabolic diseases, yearly trends, region and the gaseous pollutants, meteorological factors, public holidays, intra-week variation, average temperature, and relative humidity.	7

NOS, Newcastle–Ottawa Scale; ALP, alkaline phosphatase; ALT, alanine aminotransferase; AST, aspartate transferase; GGT, gamma-glutamyl transferase; BMI, body mass index.

**Table 2 ijerph-20-02803-t002:** Subgroup analyses of the % increase of ALP, ALT, AST, and GGT association with each 10 mg/m^3^ increase in PM2.5 concentration.

	ALP			ALT			AST			GGT		
	No. of Study	% (95% CI)	I^2^(%)	No. of Study	% (95% CI)	I^2^(%)	No. of Study	% (95% CI)	I^2^(%)	No. of Study	% (95% CI)	I^2^(%)
Region												
Asia	2	5.34 (−6.23, 16.90)	98.44	7	5.07 (0.81, 9.33)	99.59	7	4.11 (0.74, 7.48)	99.98	4	2.74 (1.09, 4.38)	52.20
Europe	0	NA	NA	1	−3.53 (−15.76, 8.70)	NA	1	2.52 (−5.85, 10.88)	NA	1	19.44 (−1.57, 40.45)	NA
USA	1	8.04 (6.79, 9.29)	NA	0	NA	NA	0	NA	NA	0	NA	NA
Sample size												
<1000	2	3.79 (−4.70, 12.27)	97.14	2	2.93 (1.03, 4.83)		2	5.84 (−1.56, 13.25)	92.39	2	2.25 (0.30, 4.20)	59.79
≥1000	1	11.18 (9.90, 12.46)	NA	6	5.94 (−1.84, 13.72)		6	3.40 (−0.28, 7.09)	99.98	3	4.73 (0.85, 8.61)	29.50
Age (years old)												
≥60	0	NA	NA	2	6.79 (−1.66, 15.23)	98.09	2	6.68 (−1.92, 15.27)	99.03	1	3.77 (1.07, 6.46)	NA
<60	2	3.79 (−4.70, 12.27)	97.14	2	2.02 (−1.56, 5.61)	74.77	2	4.47 (−4.87, 14.34)	96.14	2	3.33 (−1.43, 8.10)	67.47
Multiple ages	1	11.18 (9.90, 12.46)	NA	4	7.31 (−7.54, 22.16)	99.17	4	2.13 (−0.54, 4.79)	98.00	2	7.80 (−6.77, 22.37)	56.52
Exposure time												
Short-term	0	NA	NA	1	2.41 (0.20, 4.63)	NA	2	1.08 (−0.91, 3.06)	84.52	1	3.77 (1.07, 6.46)	NA
Long-term	3	6.28 (−0.56, 13.12)	98.26	7	5.17 (−0.19, 10.53)	99.72	6	5.02 (1.10, 8.94)	99.49	4	2.86 (0.62, 5.09)	50.30
Study design												
Prospective cohort study	2	9.61 (6.53, 12.68)	91.55	5	6.28 (−3.34, 15.90)	96.94	3	3.68 (1.15, 6.22)	74.25	2	8.00 (−5.63, 21.63)	52.43
Others	1	−0.62 (−3.21, 1.97)	NA	3	4.18 (−2.52, 10.88)	99.88	5	4.30 (−0.50, 9.11)	99.99	3	2.49 (0.58, 4.40)	49.41

NA, not applicable; CI, confidence interval; ALP, alkaline phosphatase; ALT, alanine aminotransferase; AST, aspartate transferase; GGT, gamma-glutamyl transferase.

## Data Availability

Data available in a publicly accessible repository.

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
