# Peer review of "Effect of Fine Particulate Matter Exposure on Liver Enzymes: A Systematic Review and Meta-Analysis"

_ijerph, 2023, doi:10.3390/ijerph20042803_

Round 1

Reviewer 1 Report

This manuscript presents a meta-analysis of the correlation between PM2.5 concentration and liver enzyme levels, the result showed that increments in PM2.5 exposure was significantly associated with ALT, AST, and 29 GGT. The manuscript is highly innovative and is recommended for publication after revision.

1.      Keywords are usually 4-6, please add keywords.

2.      The introductory section needs to be expanded, especially as there is less information on the current status of existing research.

3.      In section 2.1, questions about the literature search are given in more detail, such as how many documents were retrieved from each database, what the specific search formula was, when the search took place, and other details. The reproducibility of the literature search is important.

4.      The table aesthetics need to be adjusted so that one word is not split into two rows (especially in the headings section).

5.      As far as I know, research bias in meta-studies is also one of the elements that need to be studied, but I have not seen any relevant comments in the article.

6.      The formatting of references needs to be adjusted, e.g. missing page numbers.

Reviewer 2 Report

Thank you for the opportunity to review this manuscript titled, ' Effect of fine particulate matter exposure on liver enzyme: a systematic review and meta analysis'. The paper needs some major revisions before it can be accepted for publication. Below are some of my comments:

1) Line 16: Avoid using a word such as controversial. Instead you can use something like 'not robust'. 

2) In the title, it would be better to have the plural form of enzyme. i.e. enzymes because we are talking of quite a few of these liver enzymes. 

3) Line 29-30: The high levels of PM2.5 were associated with an increment or decrement of ALT, AST...etc. Please be specific. 

4) Lines 35-36: The definition for PM2.5 is not technically correct here. Not sure if all PM2.5 particles can be considered as PM2.5 particles. Please peruse through the USEPA website for more succinct definitions that are appropriate for literature type publications such as this review paper. 

5) Lines 39-40: Typically it is the ultrafine particles that have the propensity to enter the blood stream and reach the various organs of the body. Please rewrite this sentence and cite correct and appropriate articles.

6) Line 44: It should be epidemiologic studies and not epidemic studies. Check !

7) Line 45: Surgery for what? Obesity? Please clarify. 

8) Lines 86-89: The various tasks undertaken by the authors should be mentioned at the end of the manuscript under author contribution section. It should not be done in the text of the manuscript. This is the first time I have come across this. A manuscript/journal article is a team and collective effort. 

9) Lines 146-147: Please enlarge these graphs. It is difficult to decipher these especially if reading from a printout of the article. Have A, B, C, and D placed one after the other and use a whole page so that the reader does not have difficulty reading this. 

10) Lines 154-165: It this paragraph necessary. It is just replete with numbers and CI. How does it add to the overall quality of the review paper. If the authors insist on having it included in the text, then please explain your findings as well. 

11) Line 173: Statistical capacity of what? The sentence ends abruptly. 

12) Line 241: Who is the 'I' in this paragraph? 

Round 2

Reviewer 1 Report

  • Thank you for your careful revision.

Reviewer 2 Report

I perused through all the edits provided by the authors. They have incorporated all my suggestions. I, therefore, would now approve the publication of this manuscript. I do not have any further comments.